# DNA Prime and Recombinant Protein Boost Vaccination Confers Chickens with Enhanced Protection against Chicken Infectious Anemia Virus

**DOI:** 10.3390/v14102115

**Published:** 2022-09-24

**Authors:** Ling Liu, Mingrong Yin, Yang Li, Hong Su, Lichun Fang, Xiaolong Sun, Shuang Chang, Peng Zhao, Yixin Wang

**Affiliations:** 1College of Animal Science and Veterinary Medicine, Shandong Agricultural University, Tai’an 271018, China; 2China Animal Health and Epidemiology Center, Qingdao 266032, China; 3Shandong Academy of Agricultural Sciences, Jinan 250100, China

**Keywords:** chicken infectious anemia virus, DNA prime/protein boost, antibody titer, cellular immunity, protective efficacy

## Abstract

Chicken infectious anemia (CIA) is an immunosuppressive disease caused by chicken infectious anemia virus (CIAV) that poses a great threat to the poultry industry worldwide. At present, vaccination is an important way to prevent and control CIA. Apart from a CIAV-attenuated vaccine used in clinical practice, the research and development of a genetically engineered vaccine has good prospects. However, it is difficult to induce a strong protective effect with a single subunit vaccine or DNA vaccine. Therefore, the goal of this study is to develop and evaluate a DNA prime/protein boost vaccine strategy for defense against CIAV infection and spread. In this study, the recombinant proteins of CIAV VP1 and VP2 were prepared using an *Escherichia coli* (*E. coli*) expression system, and the eukaryotic expression plasmid pBud-VP1-VP2 was constructed. Subsequently, the effects of the DNA prime/protein boost strategy on antibody production and cellular immunity response were measured. The results showed that combined vaccination could induce a higher antibody titer than those of a DNA vaccine or subunit vaccine alone. In addition, spleen lymphocyte index (SI) and IL-2, IL-4, and IFN-γ levels were also significant in chickens the received the combined vaccination. To further investigate the protective effect of DNA prime/protein boost vaccination, a CIAV challenge experiment was carried out. The results showed that infection with CIAV reduced the hematocrit value (Hct) and thymus index, while vaccination recovered this reduction, and the combined immunization group was the least affected by CIAV infection. Furthermore, the CIAV viral load in the combined immunization group was the lowest, indicating that the combined immunization could provide a better protective efficacy. In conclusion, the DNA prime and recombinant protein boost vaccination can be used as an important anti-CIAV strategy, which can induce both enhanced cellular and humoral immunity responses in chickens and provide a new avenue for CIAV prevention and control.

## 1. Introduction

Chicken infectious anemia (CIA) is an immunosuppressive disease caused by chicken infectious anemia virus (CIAV), which is characterized by aplastic anemia and systemic lymphoid tissue atrophy. CIAV belongs to *Anelloviridae* and is a non-enveloped virus with icosahedral single-stranded DNA that mainly damages chicken hematopoietic cells and the T lymphocytes of immune organs [1]. The whole genome of CIAV is about 2.3 kb, containing three open-reading frames (ORFs) encoding three proteins: VP1, VP2, and VP3. The VP1 protein (51 kDa) is the only structural protein assembled into a CIAV capsid [2]. It is a major immunogenic protein that can trigger chickens to produce neutralizing antibodies. The VP2 protein (28 kDa) is a scaffold protein that helps VP1 form the correct conformation and expose its epitope [3]. The VP3 protein (13 kDa) is a non-structural protein that mainly induces apoptosis in infected chicken cells [4,5,6].

Since its first isolation in Japan, CIAV has been widely prevalent worldwide [7,8,9,10]. CIAV can be transmitted vertically or horizontally [11], is spread persistently in a chicken flock, and is difficult to eliminate [12], while bringing great economic losses to the global poultry breeding industry [13]. At present, vaccines are widely used in chicken flocks to prevent and control CIA, among which an attenuated vaccine is widely used in clinical practice. However, the CIAV-attenuated vaccine has moderate virulence and may cause damage to the immune organs of chickens under 7 weeks of age [14,15]. Therefore, breeding chickens older than 7 weeks are mainly immunized with the CIAV-attenuated vaccine to provide protection for their offspring. According to previous studies, inactivated CIAV vaccines can provide a moderate protective effect for breeding chickens [16,17]. However, since CIAV has a low titer in MDCC-MSB1 cells and chicken embryos, CIAV-inactivated vaccines have not been put into large-scale commercial use due to high production costs.

With the development of genetic-engineering technology, subunit vaccines and DNA vaccines have gained more and more attention from researchers. A subunit vaccine is a recombinant protective antigen prepared by transferring a viral protective gene into an expression vector. A subunit vaccine has strong immunogenicity, a single component, and relatively simple preparation. Subunit vaccines can induce the body to produce antibodies, but the activation of cellular immunity is limited. A DNA vaccine consists of a eukaryotic expression vector of recombinant DNA containing an antigen gene that is constructed by means of genetic recombination technology. It has the potential to simultaneously immunize multiple antigens and can stimulate the body to produce a humoral immune response and a cellular immune response at the same time [18].

Although certain progress has been made in the research and development of CIAV genetically engineered subunit vaccine and DNA vaccines [17,19,20,21,22], there are still shortcomings in using these two vaccines alone, and there are few data on viral challenge protection tests of CIAV subunit vaccines and DNA vaccines. In this study, a subunit vaccine and a DNA vaccine of CIAV are prepared, and then chickens are inoculated with the CIAV DNA vaccine and subunit vaccine through a DNA prime/protein boost strategy, and the comprehensive effect is compared with chickens immunized with these two vaccines alone. In addition, the protective effect of combined immunization with the DNA vaccine and subunit vaccine on a CIAV challenge is studied through animal experiments.

## 2. Materials and Methods

### 2.1. Expression and Purification of CIAV VP1 and VP2 Recombinant Proteins

According to the sequence of CIAV strain SD15 (GenBank ID: KX811526.1), primers were designed (Table 1), and the *VP1* and *VP2* gene fragments of SD15 were amplified through PCR. Then, the purified PCR product of *VP1* was ligated to the pET-32a (+) vector (Invitrogen, Carlsbad, USA) and digested by restriction enzymes *BamH* I and *Sal* I to construct plasmid pET32-VP1. The purified PCR product of *VP2* was ligated to the pET-28a (+) vector (Invitrogen, Carlsbad, CA, USA) and digested by restriction enzymes *BamH* I and *Eco*R I to construct plasmid pET28-VP2. To prepare VP1 and VP2 recombinant proteins, the pET32-VP1 and pET28-VP2 recombinant plasmids were transformed into BL21 *Escherichia coli* (*E. coli*)-competent cells, a single colony was cultured in LB medium, and IPTG was added with a final concentration of 1 mmol for 6 h to induce VP1 and VP2 protein expression. Then, protein purification was carried out using a Ni NTA affinity chromatography purification kit according to the instruction manual (GenScript, Nanjing, China), and 10% sodium dodecyl sulfate-polyacrylamide gel electrophoresis (SDS-PAGE) and Western blot were performed for further analysis.

### 2.2. Construction of Eukaryotic Plasmid Co-Expressing CIAV VP1 and VP2

To construct the CIAV VP1 and VP2 co-expressing plasmid, *VP1* and *VP2* fragments were amplified through PCR using the primers listed in Table 1. Then, the purified PCR product of VP2 was ligated to the pBud CE 4.1 vector (Invitrogen, Carlsbad, CA, USA) and digested by restriction enzymes *BamH* I and *Sal* I to construct plasmid pBud-VP2. After that, the purified PCR product of VP1 was ligated to pBud-VP2 and digested by restriction enzymes *Kpn* I and *Xho* I to construct recombinant plasmid pBud-VP1-VP2. To verify the successful expression of pBud-VP1-VP2 in eukaryotic cells, 2 × 10^5^ DF-1 cells were inoculated in two 24-well culture plates. Then, the pBud-VP1-VP2 plasmid was transfected into DF-1 cells using liposome 2000 (Thermo Fisher, Waltham, MA, USA) according to the instruction manual. After transfection, the cells were cultured in a 5% CO_2_ incubator at 37 ℃ for another 10 h and changed to complete DMEM medium containing 10% fetal bovine serum (FBS). The protein expression was verified by Western blot analysis and IFA assay.

### 2.3. Western Blot Analysis

Transfected DF-1 cells were collected by centrifugation, an appropriate amount of RIPA buffer lysate was added for lysis for 5 min, and the protein concentrations were determined using a BCA protein assay kit (Beyotime Biotechnology, Beijing, China). The proteins were denatured by heating, separated on 10% SDS-PAGE gels, and transferred to a nitrocellulose membrane (Millipore, Boston, MA, USA). The membranes were blocked with 5% skimmed milk in PBS containing 0.1% Tween 20 (PBST) for 1 h at room temperature and incubated with mouse anti-VP1 monoclonal antibody or rabbit anti-VP2 polyclonal antibody at 4 ℃ overnight. The blots were washed 3 times with PBST and incubated with HRP-conjugated secondary antibody (Sigma, St. Louis, MO, USA) for 1 h at room temperature. The blots were washed, and positive reactions were detected with an enhanced chemiluminescence (ECL) detection system (Beyotime Biotechnology, Beijing, China).

### 2.4. IFA Assay

Transfected DF-1 cells were fixed with 4% paraformaldehyde, and IFA was performed with mouse anti-VP1 monoclonal antibody and rabbit anti-VP2 polyclonal antibody as the primary antibodies at 4 °C overnight. Then, anti-mouse FITC-labeled secondary antibody and anti-rabbit PE-labeled secondary antibody were incubated for 1 h at room temperature. Lastly, nuclei were stained using DAPI and observed with a fluorescence microscope.

### 2.5. Evaluation of Antibody Response of Different Vaccine Immunizations

A DNA vaccine and a subunit vaccine were prepared based on the plasmid pBud-VP1-VP2 and recombinant proteins. To prepare the DNA vaccine, plasmid pBud-VP1-VP2 (1 μg/μL) was mixed with liposomes (ABclonal, Wuhan, China) at a volume ratio of 1:1. To prepare the subunit vaccine, recombinant VP1 and VP2 proteins were first mixed equivalently and then mixed with Freund’s complete adjuvant at a ratio of 1:1. A total of 60 1-day-old SPF chickens purchased from Jinan SAIS Poultry Co., LTD were randomly divided into 4 groups, with 15 chickens in each group (Table 2). Chickens in group 1 (DNA vaccine group) were immunized with 100 μg pBud-VP1-VP2 plasmid intramuscularly at both 1 day old and 14 days old. Chickens in group 2 (subunit vaccine group) were immunized with 100 μg subunit vaccine subcutaneously at both 1 day old and 14 days old. Chickens in group 3 (combined vaccine group) were immunized with 100 μg DNA vaccine at 1 day old and then immunized with 100 μg subunit vaccine at 14 days old. Chickens in group 4 (PBS control group) were injected with PBS buffer subcutaneously at both 1 day old and 14 days old. After the first immunization, serum was collected weekly, and the anti-CIAV antibody was detected using a CIAV antibody detection kit (IDEXX, Westbrook, ME, USA).

### 2.6. Lymphocyte Proliferation Assay and ELISA

Lymphocyte proliferation and concentrations of IL-2, IL-4, and IFN-γ were measured 2 weeks after the secondary immunization. Five chickens were randomly selected from each group, and the spleens of chickens were sterilely collected. Spleens were cut into pieces to prepare single-cell suspensions. Then, lymphocytes were separated using chicken lymphocyte isolation solution (Solarbio, Beijing, China) and cultured in RPMI 1640 medium with 10% FBS.

To perform the lymphocyte proliferation assay, lymphocytes were cultivated in 96-well culture plates, and 50 μL recombinant VP1 protein (5 μg/mL) was added in the medium for a total volume of 100 μL/per well. Three replicates were set in each group, and a blank control group was also set (with no recombinant VP1 protein stimulation). After 44 h, 10 μL CCK-8 solution (Beyotime Biotechnology, Beijing, China) was added to each medium and cultured for another 4 h. After that, the OD_450nm_ value was measured with a microplate reader. The results of the lymphocyte proliferation assay were expressed by value-added index (SI): SI = OD_450nm_ value of antigen stimulation wells/OD_450nm_ value of negative control wells. To detect the concentrations of IL-2, IL-4, and IFN-γ in cellular supernatant, 48 h after antigen stimulation cellular supernatants were collected, and concentrations of IL-2, IL-4, and IFN-γ were measured using an ELISA kit (Jining, Shanghai, China) according to the instruction manual. The OD_450nm_ value of each well was measured with a microplate reader, and concentrations of IL-2, IL-4, and IFN-γ were calculated using the standard curve.

### 2.7. Evaluation of Protective Efficiency of DNA Vaccine and Subunit Vaccine

A total of 75 1-day-old SPF chickens were randomly divided into 5 groups (15 per group, Table 3). Chickens in group 1 (DNA vaccine group) were immunized with the DNA vaccine and challenged with 1000 EID_50_ CIAV SD15 strain at 21 days of age intramuscularly. Chickens in group 2 (subunit vaccine group) were immunized with the subunit vaccine and challenged with 1000 EID_50_ CIAV SD15 strain at 21 days of age intramuscularly. Chickens in group 3 (combined group) were immunized with the DNA vaccine and subunit vaccine and challenged with 1000 EID_50_ CIAV SD15 strain at 21 days of age intramuscularly. Chickens in group 4 (infection control group) were challenged with 1000 EID_50_ CIAV SD15 strain at 21 days of age intramuscularly. Group 5 was the blank control group.

At the first and second weeks post virus challenge, the anticoagulants of each group were collected, and the hematocrit (Hct) was calculated. In addition, thymuses of 3 chickens in each group were collected and weighed, and the thymus index was calculated with the following formula: thymus index = thymus weight (mg)/body weight (g). At 2 weeks post virus challenge, thymuses from each group were collected, fixed with paraformaldehyde, and stained with hematoxylin and eosin (H&E staining) to perform microscopic pathological examination. At the first and second weeks post challenge, the blood of chickens in each group was collected, and the viral load was detected through absolute quantitative real-time PCR using a SYBR Green I qPCR kit (Takara, Dalian, China) according to the instruction manual. The primers used are listed in Table 1, and viral loads were calculated using an equation generated according to the standard curve.

### 2.8. Statistical Analysis

The results are presented as the means ± the SEMs. All statistical analyses were performed using SPSS statistical software for Windows, version 17.0 (SPSS Inc., Chicago, IL, USA). Statistical comparisons were made using Student’s *t*-test.

## 3. Results

### 3.1. Expression of Recombinant VP1 and VP2 Proteins by E. coli Cells

Prokaryotic expression plasmids pET32-VP1 and pET28-VP2 were constructed and transformed into E. coli cells for induced expression. The results of SDS-PAGE showed that the recombinant VP1 fusion protein was about 74 kDa in size, and the VP2 protein was about 26 kDa in size; VP1 and VP2 recombinant proteins existed both in the supernatant fluid and the inclusion body. Subsequently, high-purity purified protein was obtained with a Ni NTA protein purification kit (Figure 1). The results of the Western blot analysis showed that both bands could be recognized by mouse anti-VP1 monoclonal antibody or rabbit anti-VP2 polyclonal antibody, indicating that the VP1 and VP2 proteins were accurately expressed (Figure 1).

### 3.2. Construction of CIAV VP1 and VP2 Eukaryotic Co-Expression Plasmid

The VP1 and VP2 gene sequences of the CIAV SD15 strain were amplified and ligated into pBud CE 4.1 vector, respectively, to construct VP1 and VP2 eukaryotic co-expression plasmid pBud-VP1-VP2. The plasmid pBud-VP1-VP2 could be digested by a restriction enzyme, indicating that the plasmid was constructed correctly (Figure 2A,B). The plasmid pBud-VP1-VP2 was transfected into DF-1 cells, and the cells were collected 36 h later for Western blot analysis. The results showed that two bands could be detected with anti-His monoclonal antibody, one at about 54 kDa (VP1) and the other at about 24 kDa (VP2), indicating that the recombinant plasmid pBud-VP1-VP2 could be successfully expressed in DF-1 cells (Figure 2C). Furthermore, transfected DF-1 cells were used for IFA verification. The results showed that the cells transfected with plasmid pBud-VP1-VP2 could be recognized by anti-VP1 antibody (green) and anti-VP2 antibody (red) (Figure 3).

### 3.3. Determination of Antibody Titer Induced by Different Vaccine Immunization

Serums were collected weekly after the first immunization for antibody monitoring. The results showed that no antibody was produced in the empty vector control group, while the antibody levels of the other three groups began to rise after the first immunization and increased significantly after the second immunization (Figure 4). Moreover, the antibody levels induced by the combined immunization of DNA vaccine and subunit vaccine 2 weeks after the second immunization were higher than those of both the DNA vaccine group and the subunit vaccine group. This indicated that the DNA vaccine and subunit vaccine prepared in this study could stimulate the humoral immune response, and the DNA prime/protein boost vaccination could induce the highest CIAV antibody titer.

### 3.4. Effect of Different Vaccine Immunization on Cellular Immune Function

To determine the effect of vaccine immunization on the cellular immune function of chickens, the proliferation of spleen lymphocytes and cytokine levels were detected through a lymphocyte proliferation test and an ELISA assay in vitro. The results showed that the lymphocyte proliferation levels in all the vaccinated groups were higher than that of the control group, and the SI index of chickens immunized with the DNA vaccine combined with the subunit vaccine was significantly higher than those of other immunized groups (*p* < 0.05, Figure 5A). In addition, the levels of IL-2, IL-4, and IFN-γ secreted by cultured spleen lymphocytes of chickens immunized with the DNA vaccine combined with the subunit vaccine were the highest when compared with other groups (Figure 5B).

### 3.5. Determination of Hct and Thymus Index of Chickens after CIAV Challenge

To determine the protective effect of the DNA prime/protein boost strategy, vaccinated chickens were challenged with CIAV after the secondary immunization, and the Hct and thymus index of chickens were detected. The results showed that the Hct value of chickens in the CIAV challenge group significantly decreased, while the Hct value of vaccinated chickens as less-affected. Among them, the Hct value of chickens immunized with the DNA vaccine combined with the subunit vaccine was the least affected by CIAV challenge (Table 4). The thymus index of chickens immunized with the DNA vaccine combined with the subunit vaccine was the least affected by CIAV challenge when compared with the other groups (Table 5).

### 3.6. Histopathological Observation of Thymus after CIAV Challenge

The thymus tissue of chickens in each group was collected to prepare pathological sections at 2 weeks post CIAV challenge. The results showed that obvious cortical degeneration could be observed in chickens of the DNA vaccine group (Figure 6A), while the thymic structure was relatively integrated in chickens of the subunit vaccine group (Figure 6B). Compared with the above two groups, the thymuses of the chickens in the combined vaccine group showed basically normal histrionic structure (Figure 6C). In the CIAV challenge group, hemorrhage and degeneration of the thymic cortex could be observed, and the remaining thymic medulla was visible (Figure 6D). In contrary, the thymic cortex and medulla of the control group were clearly demarcated, and the lymphocyte development was normal in chickens of the control group (Figure 6E).

### 3.7. Determination of Viral Load in Plasma after CIAV Challenge

Viral load in plasma was detected at 2 weeks post CIAV challenge, and the total positive rate of viral DNA and the mortality rate of each group were counted. The results showed that, except for the control group, viremia could be detected in all other groups after the challenge (Table 6). The viral load of the vaccination group was significantly lower than that of the challenge group. Among them, the chickens of the combined vaccine group had the lowest viral load, and the positive rate of viral DNA was the lowest, indicating that the DNA prime/protein boost vaccination could play a better protective role for chickens.

## 4. Discussion

In view of the serious harm of CIAV to global poultry production, especially broiler production, the poultry industry has paid more and more attention to the prevention and control of this disease. At present, therapeutic drugs for CIA are in the research stage [23,24], so vaccine immunization is the main way to prevent CIAV infection. In some regions, Cux-1- or 26p4-strain-attenuated vaccines are used to immunize breeding chickens to protect their offspring from CIAV infection. However, attenuated CIAV vaccines have strong virulence and are only suitable for chickens over 7 weeks old; in addition, there is a safety risk of virulence reversion for attenuated CIAV vaccines [15]. CIAV-inactivated vaccines can protect offspring from virulent attack [16,17]. However, due to the low virus titer of CIAV in chicken embryos or MDCC-MSB1 cells, the production cost of CIAV-inactivated vaccines is greatly increased, which makes it difficult to popularize inactivated vaccines in large quantities. Therefore, it is of great significance to develop a safe and effective vaccine to control CIAV.

In recent years, the development of genetically engineered vaccines has become the focus of CIAV vaccine research. Studies have shown that the antibody response produced by the VP1 protein alone is weak, while the co-expression of the VP1 and VP2 proteins has stronger immunogenicity [22,25]. At present, a variety of expression systems are used to express VP1 alone or the VP1-VP2 protein to prepare CIAV subunit vaccines, such as *E. coli* expression systems, yeast expression systems, silkworm, tobacco, etc. [25,26,27]. Due to the weak immunogenicity of subunit vaccines, adjuvants are often added to enhance the immune effect, such as oil emulsion adjuvant and CpG adjuvant [21]. Researchers have also tried to integrate different genes, such as *IFN-γ* and the *VP22* gene of Marek’s disease virus, to the *VP1* sequence to enhance the immunogenicity of subunit vaccines [20,28]. In this study, full-length recombinant VP1 protein and VP2 protein of CIAV were respectively expressed in E. coli, and the temperature, induction time, and IPTG concentration were adjusted to realize the soluble expressions of VP1 and VP2 proteins. It should be pointed out that we tried to ligate the VP1 gene to the pET28-a(+) vector first; however, the protein was not successfully expressed. Therefore, the *VP1* gene was cloned into the pET32-a(+) vector, and recombinant VP1 with a tag was successfully expressed.

Other than subunit vaccines, DNA vaccines have attracted extensive attention as a new type of vaccine in recent years [29]. A DNA vaccine can express a target antigen in the host cell in vivo, which can directly bind with MHC-I and II molecules to trigger specific cellular and humoral immunity. DNA vaccines not only have the effectiveness to induce a wide range of immune responses, but also can induce cellular immune responses, and there is no risk of virulence reversion in vivo. In this study, full-length *VP1* and *VP2* genes were cloned into pBud CE 4.1 plasmid to construct co-expression eukaryotic plasmid. We placed the full-length *VP1* fragment under the CMV promoter and the full-length *VP2* fragment under the EF-1 α promoter. To verify its expression, we transfected it into DF-1 cells and tested it with mouse anti-VP1 antibody and rabbit anti-VP2 antibody, respectively. The results showed that the plasmid could be expressed successfully.

Theoretically, subunit vaccines have advantages in inducing a humoral immune response, but they have limitations in inducing comprehensive cellular immunity. DNA vaccines have a certain ability to induce a high level of cellular immunity, but the protective effect on virus challenge is still insufficient. To achieve a better protective efficiency, researchers use a DNA vaccine for primary immunization and then use a subunit vaccine to enhance the immune effect [30]. This DNA prime/recombinant protein boost vaccination strategy can produce a more ideal immune effect than that of a single vaccine, and this strategy has been used in the research of multiple vaccines [31,32,33]. It is generally considered that the number of memory T cells produced by the prime/boost strategy is higher than that of a single vaccine, and it has a better effect in inducing cellular immune response. However, the immune mechanism of the prime/boost strategy needs to be further studied.

In order to investigate the application of the DNA prime/protein boost strategy in the immune protection of CIAV, we used a DNA vaccine for primary immunization and a subunit vaccine for booster immunization in this study and evaluated the immune effect of this strategy in SPF chickens. The results of showed that immunization with the two vaccines separately could induce antibody titers in chickens, and the antibody titer induced by the subunit vaccine was higher than that of the DNA vaccine. When the two vaccines were combined for vaccination, a higher level of antibody response could be induced. Subsequently, the effect of DNA prime/protein boost vaccination on cellular immune function was measured. The results showed that the DNA vaccine could induce a higher level of IFN-γ than that of the subunit vaccine, while the subunit vaccine could induce a higher level of IL-4 than that of the DNA vaccine. It is generally considered that IL-4 is secreted by Th2 CD4+ lymphocytes, which can promote the production of IgG1 antibody by effector B cells, while IFN-γ is secreted by Th1 CD4+ lymphocytes, which can promote the production of IgG2a antibody. Therefore, the DNA prime/protein boost strategy could simultaneously improve the immune responses of Th1 and Th2, as well as improve the humoral and cellular immunity levels of SPF chickens.

In order to further investigate the protective effect of DNA prime/protein boost vaccination, a CIAV challenge protection test was carried out. Because chickens of older age are not sensitive to CIAV infection and do not show obvious pathological changes, there is a lack of sufficient indicators for the evaluation of vaccine protection, which is an important factor limiting the research of CIAV vaccine at present. Therefore, we mainly evaluated the Hct value, thymus index, and viral load in this study. The results showed that the DNA vaccine alone had a poor protective effect on the CIAV challenge, while the subunit vaccine alone had a certain protective effect better than that of the DNA vaccine, and the combined immunization had the best protective effect on the CIAV challenge. These results demonstrated that DNA prime/protein boost vaccination could induce chickens to produce higher levels of antibody response and cellular immune response, which showed a good application prospect for CIAV prevention and control.

In summary, CIAV DNA vaccines and subunit vaccines were prepared in this study, and a DNA prime/protein boost vaccination strategy was successfully established. This immunization strategy could provide a better immune protection effect and has a certain significance for CIAV vaccine research in the future.

## Figures and Tables

**Figure 1 viruses-14-02115-f001:**
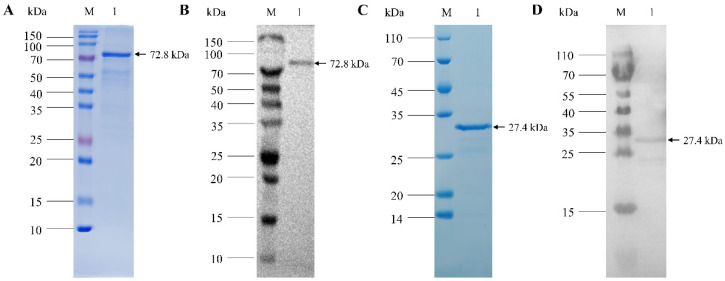
Purification and Western blot analysis of CIAV recombinant VP1 and VP2 proteins. (**A**): SDS-PAGE of purified recombinant CIAV VP1 protein. (**B**): Western blot analysis of CIAV VP1 protein using anti-VP1 monoclonal antibody. (**C**): SDS-PAGE of purified recombinant CIAV VP2 protein. (**D**): Western blot analysis of CIAV VP2 protein using anti-VP2 polyclonal antibody.

**Figure 2 viruses-14-02115-f002:**
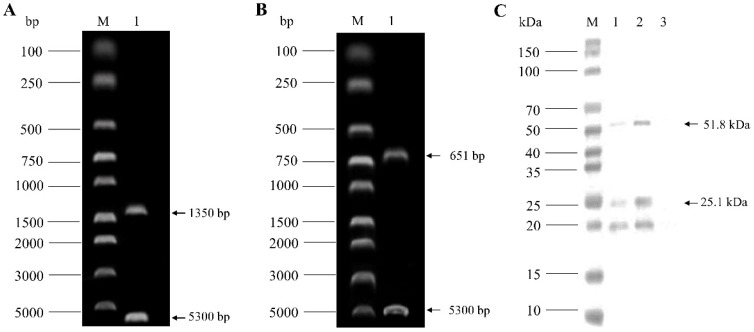
Identification of pBud-VP1-VP2 plasmid by enzyme digestion and Western blot analysis. (**A**): Enzyme digestion of pBud-VP1-VP2 by *Kpn* I and *Xho* I. (**B**): Enzyme digestion of pBud-VP1-VP2 by *Sal* I and *BamH* I. (**C**): Western blot analysis of total protein of DF-1 transfected with pBud-VP1-VP2 using anti-His monoclonal antibody.

**Figure 3 viruses-14-02115-f003:**
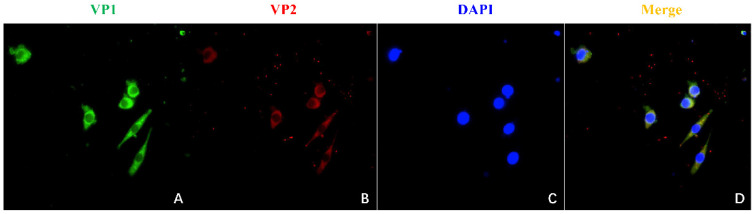
Identification of pBud-VP1-VP2 expression by IFA. (**A**): IFA detection with mouse anti-VP1 antibody. (**B**): IFA detection with rabbit anti-VP2 antibody. (**C**): Staining with DAPI. (**D**): Merge of (**A**–**C**).

**Figure 4 viruses-14-02115-f004:**
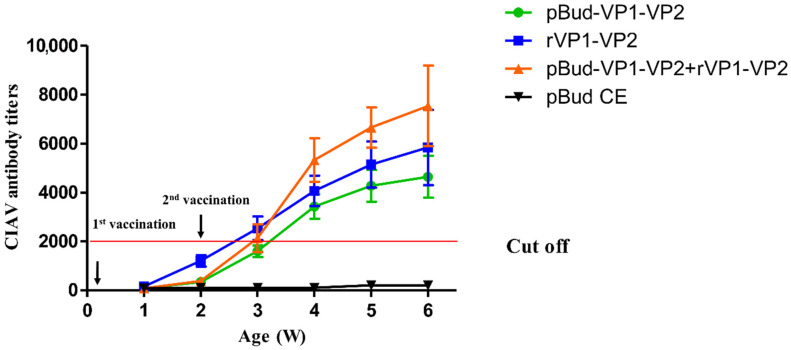
Antibody titer determination after vaccination.

**Figure 5 viruses-14-02115-f005:**
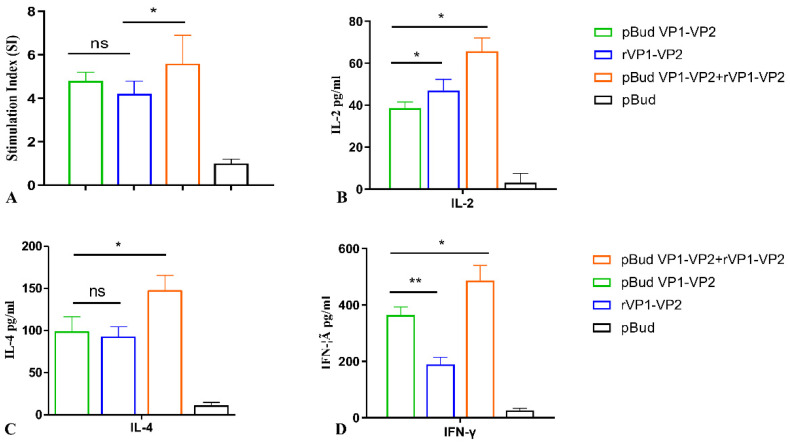
Detection of SI index of splenic lymphocytes and expression levels of IL-2, IL6, and IFN-γ in vitro. (**A**): SI index of splenic lymphocytes stimulated with VP1 protein in vitro. (**B**): Expression levels of IL-2 of spleen lymphocytes stimulated with VP1 protein in vitro. (**C**): Expression levels of IL-6 of spleen lymphocytes stimulated with VP1 protein in vitro. (**D**): Expression levels of IFN-γ of spleen lymphocytes stimulated with VP1 protein in vitro. Differences were considered significant when *p* ≤ 0.05 (*) and highly significant when *p* ≤ 0.01 (**); the error bars represent the SEM; ns, no significant difference (*p* > 0.05).

**Figure 6 viruses-14-02115-f006:**
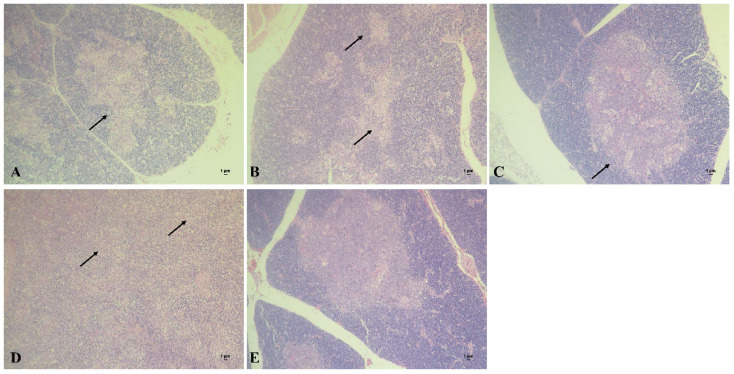
Section observation of chicken thymuses in different groups after CIAV challenge. (**A**): Chicken thymus of DNA vaccine group (100×). (**B**): Chicken thymus of subunit vaccine group (100×). (**C**): Chicken thymus of combined vaccine group (100×). (**D**): Chicken thymus of CIAV infection control group (100×). (**E**): Chicken thymus of blank control group (100×).

**Table 1 viruses-14-02115-t001:** Primer sequence used in this study.

NO	Primers	Sequences (5′-3′)
1	pET-VP1-F	GGATCCATGGCTCGTCGTGCTCGTA
2	pET-VP1-R	GTCGACCGGCGGAGAACCCCAGTA
3	pET-VP2-F	GGATCCATGCACGGTAACGGTGGTCA
4	pET-VP2-R	GAATTCAACGATACGAACCGGAGCC
5	pBud-VP1-F	GGTACCGCCACCATGGCAAGACGAGCTCGCAG
6	pBud-VP1-R	CTCGAGTCAATGGTGATGGTGATGATGGGGGGGC
7	pBud-VP2-F	GTCGACAGCCACCATGCACGGGAACGGCGGAC
8	pBud-VP2-R	GGATCCTCAATGGTGATGGTGATGATGCACTATA

^1^ Primers 1–4 were used to construct prokaryotic expression plasmids pET32-VP1 and pET28-VP2; primers 5–8 were used to construct eukaryotic expression plasmid pBud-VP1-VP2. Underline represents restriction enzyme recognition sites.

**Table 2 viruses-14-02115-t002:** Groups of different vaccine immunization tests.

Group	Number	Immunization Treatment
DNA vaccine group	15	Immunized with DNA vaccine at 1 day and 14 days of age (100 μg)
Subunit vaccine group	15	Immunized with subunit vaccine at 1 day and 14 days of age (100 μg)
Combined vaccine group	15	Immunized with DNA vaccine at 1 day of age (100 μg) and immunized with subunit vaccine at 14 days of age (100 μg)
PBS control group	15	Injected with PBS buffer at 1 day and 14 days of age

**Table 3 viruses-14-02115-t003:** Groups of vaccine protective tests.

Group	Number	Immunization Treatment	CIAV Challenge
DNA vaccine group	15	Immunized with DNA vaccine at 1 day and 14 days of age (100 μg)	Challenged with 1000 EID_50_ CIAV at 21 days of age
Subunit vaccine group	15	Immunized with subunit vaccine at 1 day and 14 days of age (100 μg)	Challenged with 1000 EID_50_ CIAV at 21 days of age
Combined vaccine group	15	Immunized with DNA vaccine at 1 day of age (100 μg) and immunized with subunit vaccine at 14 days of age (100 μg)	Challenged with 1000 EID_50_ CIAV at 21 days of age
Infection control group	15	-	Challenged with 1000 EID_50_ CIAV at 21 days of age
Blank control group	15	-	-

**Table 4 viruses-14-02115-t004:** Hematocrit value (Hct) after CIAV challenge.

Group	1 w Post CIAV Challenge	2 w Post CIAV Challenge
DNA vaccine group	0.38 ± 0.04 ^b^	0.33 ± 0.05 ^b^
Subunit vaccine group	0.40 ± 0.02 ^b^	0.34 ± 0.03 ^b^
Combined vaccine group	0.42 ± 0.03 ^a^	0.38 ± 0.02 ^a^
Infection control group	0.38 ± 0.04 ^b^	0.29 ± 0.02 ^c^
Blank control group	0.46 ± 0.03 ^a^	0.41 ± 0.02 ^a^

Different lowercase letters represent statistically significant differences between the groups (*p* < 0.05).

**Table 5 viruses-14-02115-t005:** Thymus index after CIAV challenge.

Group	1 w Post CIAV Challenge	2 w Post CIAV Challenge
DNA vaccine group	3.04 ± 0.42 ^b^	3.28 ± 0.68 ^b^
Subunit vaccine group	3.40 ± 0.94 ^b^	3.29 ± 1.21 ^b^
Combined vaccine group	3.45 ± 1.74 ^b^	3.96 ± 0.68 ^a^
Infection control group	2.85 ± 1.25 ^c^	2.83 ± 0.55 ^c^
Blank control group	3.93 ± 1.36 ^a^	4.01 ± 0.92 ^a^

Different lowercase letters represent statistically significant differences between the groups (*p* < 0.05).

**Table 6 viruses-14-02115-t006:** Viral load, positive rate of CIAV, and mortality rate after CIAV challenge.

Group	Viral Load (1 w Post Challenge, log10)	Viral Load (2 w Post Challenge, log10)	Positive Rate	Mortality Rate
DNA vaccine group	4.41 ± 0.47 ^b^	3.83 ± 0.72 ^b^	8/15	1/15
Subunit vaccine group	3.65 ± 0.54 ^c^	2.48 ± 0.41 ^b^	6/15	0/15
Combined vaccine group	2.33 ± 0.40 ^d^	1.65 ± 0.25 ^c^	4/15	0/15
Infection control group	5.93 ± 0.66 ^a^	4.85 ± 0.47 ^a^	15/15	3/15
Blank control group	-	-	0/15	0/15

Different lowercase letters represent statistically significant differences between the groups (*p* < 0.05). Data of viral loads in this table are the average values of viral-positive chickens.

## Data Availability

No new data were created or analyzed in this study. Data sharing is not applicable to this article.

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
