# Peer review of "DNA Prime and Recombinant Protein Boost Vaccination Confers Chickens with Enhanced Protection against Chicken Infectious Anemia Virus"

_viruses, 2022, doi:10.3390/v14102115_

Round 1

Reviewer 1 Report

In the manuscript, Ling Liu et al. developed and evaluated a DNA prime/protein boost vaccine strategy for defending CIAV infection and spreading. They concluded that DNA-prime and recombinant protein boost vaccination could be used as an important way of anti-CIAV strategy, which could induce both enhanced cellular and humoral immunity responses in chickens. However, there are several major questions should be considered: 

1. Why did you immnue DNA vaccine first, and then subunit vaccine second? Why did you not design to immune subunit vaccine first and then DNA vaccine second at the same time?

2. The expressed protein and DNA vector can be use as vaccine diretly? 

3. In table 6, the viral loads are all chickens or viral positive chickens? 

Minor concerns:

1.  Line 39: Please check "CIAV belongs to circoviridae" . CAV is redesigned to Anelloviridae.

2. Line 53: However, CIAV attenuated vaccine has strong virulence and may cause damage to immune organs of chickens after vaccination“. Please check the accuracy of this sentence description.

3. Please notice "control group" in table 2 and 3 is not precise. 

Author Response

Dear reviewer,

Thank you for your letter and for the reviewers’ comments concerning our manuscript. Firstly, we would like to thank the reviewers and the editor for the constructive comments and suggestions. We have studied comments carefully and have substantially revised our manuscript which we hope meet with approval. The main corrections in the paper and the responds to the reviewer’s comments are as following.

  1. Why did you immnue DNA vaccine first, and then subunit vaccine second? Why did you not design to immune subunit vaccine first and then DNA vaccine second at the same time?

Reply: Thanks for your suggestion. In fact, we had also considered your questions when we design this experiment. As we know, the prime/boost immunization strategy is to use one vaccine for initial immunization, and then use another heterogeneous vaccine for booster immunization. The major advantage of prime/boost strategy is that it can effectively induce cellular immune response, certainly with the humoral immune response. Although the precise mechanism of prime/boost strategy to promote immune response is not fully elucidated, it is generally agreed that exogenous protein expressed by plasmid DNA in the host cells can be presented by MHC class I pathway, and then stimulate T cell immune response. During this process, humoral immunity can also be stimulated, although it is relatively weak. After a period of time, recombinant protein is used to strengthen the humoral immunity and induce strong humoral immunity response. In contrary, if the protein primary immunization is used first, the effect of stimulating cellular immunity may be weakened. We also consulted relevant research papers. From published reference papers, this DNA-prime/protein boost strategy has been widely used. Therefore, we also adopted this approach, namely DNA was immunized first, and then protein was immunized.

  1. The expressed protein and DNA vector can be use as vaccine diretly? 

Reply: In our experiment, the expressed recombinant protein was mixed with Freund's complete adjuvant (1:1). DNA vaccine is mixed with liposome (ABclonal, Wuhan, China), and each chicken is inoculated with 100 μL plasmid+100 μL Liposome. We have modified it in the revised manuscript. (Line 133-137)

  1. In table 6, the viral loads are all chickens or viral positive chickens? 

Reply: The viral load in Table 6 is the data of viral positive chickens. We have modified it in the revised manuscript. (Line 308)

  1. Line 39: Please check "CIAV belongs to circoviridae". CAV is redesigned to Anelloviridae.

Reply: Thanks for your suggestion. We have corrected it in the revised manuscript. (Line 40)

  1. Line 53: However, CIAV attenuated vaccine has strong virulence and may cause damage to immune organs of chickens after vaccination. Please check the accuracy of this sentence description.

Reply: Thanks for your suggestion. We have modified it in the revised manuscript as following “However, CIAV attenuated vaccine has moderate virulence and may cause damage to immune organs of chickens under 7 weeks of age”. (Line 54-55)

  1. Please notice "control group" in table 2 and 3 is not precise. 

Reply: Thanks for your suggestions. “Control group” in Table 2 has been revised as “PBS control group”, “Control group” in Table 3 has been revised as “Blank control group”, and “Infection group” in Table 3 has been revised as “Infection control group”.

Finally, we appreciate very much for your time in editing our manuscript and the valuable suggestions and comments. I am looking forward to hearing from your decision when it is made.

Yixin Wang

Sep, 18, 2022

Reviewer 2 Report

The paper reported a new stragery for control CIAV infection in the chickens. The results were instersting and novelty for the prevention of the disease. However, there were some erros in the manuscript.

1. Line 19The abbreviation “E. coli” first emerged and it should be given the full name.

2. Line 74: “firstly” should be deleted.

3. Line 90: A space should be added between 1 and mmol.

4. Line 108: It should be 5% CO2.

5. Line 116: The abbreviation SDS-PAGE has been shown in the line 92 of the paper.

6. Line 119: The temperature “” should be Times New Roman.

7. Line 135-136: Which route was for the subunit vaccine?

8. The ethical approval of animal experiments should be given.

9. The lymphocytes were stimulated with the recombinant VP1 protein. Why not used the VP2 protein

10. The OD450 should be OD450nm.

11. Which route was for the viral inoculation?

12. Line192-195: The sentences “The VP1 and VP2 gene fragments of CIAV SD15 strain were amplified, and purified VP1 fragment was ligated into pET32-a(+) vector to construct pET32-VP1 plasmid; the purified VP2 fragment was ligated into pET28-a(+) vector to construct pET28-VP2 plasmid. Then, pET32-VP1 and pET28-VP2 were transformed into BL21 E. coli cells respectively, and IPTG was added to induce protein expression.” were the descriptions of methods. Then, they should be deleted.

13. “western blot” or “western blotting”? It should be the same in the manuscript.

14. “Figure A2, Figure B2…….”? They should be Figure 2A….

15. The statistical analysis of the table 4, table 5 and table 6 should be more accurate.

16. For the figure 6, the pathological changes should be given using the arrows.

Author Response

Dear reviewer,

Thank you for your letter and for the reviewers’ comments concerning our manuscript. Firstly, we would like to thank the reviewers and the editor for the constructive comments and suggestions. We have studied comments carefully and have substantially revised our manuscript which we hope meet with approval. The main corrections in the paper and the responds to the reviewer’s comments are as following.

  1. Line 19The abbreviation “E. coli” first emerged and it should be given the full name.

Reply: Thanks for your suggestions. We have modified it in the revised manuscript. (Line 19, 89)

  1. Line 74: “firstly” should be deleted.

Reply: Thanks for your suggestions. We have deleted “firstly” in the revised manuscript. (Line 75)

  1. Line 90: A space should be added between 1 and mmol.

Reply: Thanks for your suggestions. We have modified it in the revised manuscript. (Line 91)

  1. Line 108: It should be 5% CO2.

Reply: Thanks for your suggestions. We have modified it in the revised manuscript. (Line 110)

  1. Line 116: The abbreviation SDS-PAGE has been shown in the line 92 of the paper.

Reply: Thanks for your suggestions. We have modified it in the revised manuscript. (Line 93-94)

  1. Line 119: The temperature “℃” should be Times New Roman.

Reply: Thanks for your suggestions. We have modified it in the revised manuscript. (Line 110)

  1. Line 135-136: Which route was for the subunit vaccine?

Reply: In our experiment, the expressed recombinant protein was mixed with Freund's complete adjuvant (1:1), and immunized subcutaneously. We have modified it in the revised manuscript. (Line 133-137)

  1. The ethical approval of animal experiments should be given.

Reply: The ethical approval is automatically generated when we uploaded this article, and the full text of the ethical approval can be checked in the “Institutional Review Board Statement” section. (Line 403-407)

  1. The lymphocytes were stimulated with the recombinant VP1 protein. Why not used the VP2 protein

Reply: It is generally considered that VP1 contains most of epitopes of CIAV, and be regarded as the main protective antigen of CIAV. Therefore, we use recombinant VP1 protein to stimulate the lymphocytes in our experiment.

  1. The OD450 should be OD450nm.

Reply: Thanks for your suggestions. We have modified it in the revised manuscript. (Line 162)

  1. Which route was for the viral inoculation?

Reply: Thanks for your suggestions. Chickens was injected with CIAV in an intramuscular way in our experiment. We have modified it in the revised manuscript. (Line 142, 145)

  1. Line192-195: The sentences “The VP1 and VP2 gene fragments of CIAV SD15 strain were amplified, and purified VP1 fragment was ligated into pET32-a(+) vector to construct pET32-VP1 plasmid; the purified VP2 fragment was ligated into pET28-a(+) vector to construct pET28-VP2 plasmid. Then, pET32-VP1 and pET28-VP2 were transformed into BL21 E. coli cells respectively, and IPTG was added to induce protein expression.” were the descriptions of methods. Then, they should be deleted.

Reply: Thanks for your suggestions. Those descriptions had been simplified in the revised manuscript. (Line 199-200)

  1. “western blot” or “western blotting”? It should be the same in the manuscript.

Reply: Thanks for your suggestions. The terms have been unified as “western blot”. (Line 210, 212)

  1. “Figure A2, Figure B2…….”? They should be Figure 2A….

Reply: Thanks for your suggestions. We have modified this issues in the revised manuscript.

  1. The statistical analysis of the table 4, table 5 and table 6 should be more accurate.

Reply: Thanks for your suggestions. The statistical analysis in Table 4, 5, and 6 was carefully checked, and lowercase letters were superscripted in the revised manuscript.

  1. For the figure 6, the pathological changes should be given using the arrows.

Reply: Thanks for your suggestions. The pathological changes had been pointed by the arrows in the revised manuscript.

Finally, we appreciate very much for your time in editing our manuscript and the valuable suggestions and comments. I am looking forward to hearing from your decision when it is made.

Yixin Wang

Sep, 18, 2022

Round 2

Reviewer 1 Report

Language should be improved again.